# Neutralizing Antibodies against SARS-CoV-2, Anti-Ad5 Antibodies, and Reactogenicity in Response to Ad5-nCoV (CanSino Biologics) Vaccine in Individuals with and without Prior SARS-CoV-2

**DOI:** 10.3390/vaccines9091047

**Published:** 2021-09-20

**Authors:** Jorge Hernández-Bello, José Javier Morales-Núñez, Andrea Carolina Machado-Sulbarán, Saúl Alberto Díaz-Pérez, Paola Carolina Torres-Hernández, Paulina Balcázar-Félix, Jesús Alberto Gutiérrez-Brito, José Alvaro Lomelí-Nieto, José Francisco Muñoz-Valle

**Affiliations:** 1Institute of Research in Biomedical Sciences, University Center of Health Sciences (CUCS), University of Guadalajara, Guadalajara 44340, Mexico; jorge89_5@hotmail.com (J.H.-B.); pepejavis_15@hotmail.com (J.J.M.-N.); saulalberto95@outlook.com (S.A.D.-P.); vabj_94@hotmail.com (J.A.G.-B.); loman_an@hotmail.com (J.A.L.-N.); 2Institute for Research on Cancer in Childhood and Adolescence, University Center of Health Sciences (CUCS), University of Guadalajara, Guadalajara 44340, Mexico; andrea.machado5223@academicos.udg.mx; 3Immunology Laboratory, University Center of Health Sciences (CUCS), University of Guadalajara, Guadalajara 44340, Mexico; carolina.torres.h@hotmail.com (P.C.T.-H.); paufe97@gmail.com (P.B.-F.)

**Keywords:** COVID-19, SARS-CoV-2, AD5-nCOV, neutralizing antibodies, vaccine, reactogenicity

## Abstract

This is the first study outside of clinical trials (phase I–III) evaluating the ability of the Ad5-nCoV vaccine to generate neutralizing antibodies and the factors associated with optimal or suboptimal response. In a longitudinal assay, 346 people (117 with prior COVID-19 and 229 without prior COVID-19) vaccinated with Ad5-nCoV were recruited. The percentage of neutralizing antibodies against SARS-CoV-2 (Surrogate Virus Neutralization Test) and antibodies against Ad5 (ADV-Ad5 IgG ELISA) were quantified pre and post-vaccination effects. The Ad5-nCoV vaccine induces higher neutralizing antibodies percentage in individuals with prior COVID-19 than those without prior COVID-19 (median [IQR]: 98% [97–98.1] vs. 72% [54–90], respectively; *p* < 0.0001). Furthermore, a natural infection (before vaccination) induces more neutralizing antibodies percentage than immunized individuals without prior COVID-19 (*p* < 0.01). No patient had vaccine-severe adverse effects. The age, antidepressant, and immunosuppressive treatments, reactogenicity, and history of COVID-19 are associated with impaired antibody production. The anti-Ad5 antibodies increased after 21 days of post-vaccination in all groups (*p* < 0.01). We recommend the application of a booster dose of Ad5-nCoV, especially for those individuals without previous COVID-19 infection. Finally, the induction of anti-Ad5 antibodies after vaccination should be considered if a booster with the same vaccine is planned.

## 1. Introduction

Since December 2019, coronavirus disease 2019 (COVID-19) has caused more than four million deaths worldwide [1], mainly in older adults and patients with underlying diseases such as diabetes, hypertension, obesity, lung diseases, and cardiovascular diseases [2].

Innate immunity is the first line of defense against pathogens and induces adaptive immunity, which generates immunological memory [3]. Cellular and humoral memory are relevant mechanisms to control virus replication and dissemination and prevent future reinfections. B-lymphocytes participate in the IgM and IgG antibodies production, which can mediate a neutralizing role to directly interfere with SARS-CoV-2 internalization in the epithelial cells [3,4]. The neutralizing antibodies rapidly appear in serum after SARS-CoV-2 infection and vaccination and are maintained for several months [5].

The evolution in technology and sequencing of the SARS-CoV-2 genome allowed the accelerated development of various vaccines types [6,7], including viral vector-based vaccines, mRNA and DNA vaccines, subunit vaccines, nanoparticle-based vaccines, and inactivated-whole virus vaccines [8]. Several of these vaccine types are being approved in record time to minimize the high lethality of this virus.

CanSino Biologics Inc., in collaboration with the Beijing Institute of Biotechnology, developed a non-replicating viral vector vaccine against SARS-CoV-2 called Adenovirus Type 5 Vector (Ad5-nCoV), which is currently undergoing Phase III clinical evaluation [9]. This vaccine has an emergency use authorization only in eight countries, including Argentina, Chile, China, Ecuador, Hungary, Malaysia, Mexico, and Pakistan [10].

The Ad5-nCoV vaccine has 5 × 10^10^ viral particles per mL and is currently applied in a single dose. According to a previous report, a single intramuscular immunization produces significant immune responses in most of recipients and induces the generation of neutralizing antibodies that responds to live SARS-CoV-2 [11].

The Ad5-nCoV encodes the full spike protein of SARS-CoV-2 and has shown enough immunogenicity in human clinical trials. However, at least one adverse reaction within the first seven days after the vaccination has been reported, including pain at the injection site, fever, headache, fatigue, and myalgia [11,12]. Comparisons between diverse vaccine types, evidence that Ad5–nCoV is less effective than other mRNA vaccines; however, it is more effective to produce neutralizing antibodies than other viral vector vaccines [13].

A disadvantage of vector-based vaccines is that innate immunity can control both vector and vaccine and could not discriminate between the two genomes. Therefore, adaptive immunity can be induced by the vector and the vaccine antigen, generating immunity to both [14].

Ad5-nCoV has a viral vector from Adenovirus 5 (Ad5), which in natural conditions causes respiratory infections, mainly in the pediatric population. Epidemiological studies have attributed to this virus 5–10% of all febrile illnesses in infants, which are related to respiratory tract symptoms [15]. In adults, there are different seroprevalence of anti-Ad5 antibodies around the world, e.g., in the United States (U.S.) the prevalence of these antibodies is 57% [16] whereas in Gambia it is 84% [17].

In Mexico, the prevalence of Ad5 infection in adults and infants has not been studied thoroughly; however, it could be similar to the U.S. percentage. Therefore, it is unknown if a history of Ad5 infection could influence the efficacy of the current Mexican vaccine strategy, which includes the Ad5-nCoV vaccine.

This study aimed to determine the vaccine-associated side effects, the generation of neutralizing antibodies against SARS-CoV-2 and anti-Ad5 antibodies in a group of Mexican patients immunized with the Ad5-nCoV vaccine with and without a history of COVID-19.

## 2. Materials and Methods

### 2.1. Subjects and Sample Collection

We included 346 individuals (teachers and administrative staff of public schools) from Guadalajara, Mexico, who had been vaccinated with Ad5-nCoV (CanSino Biologics Inc). All subjects were recruited in the Centro Universitario de Ciencias de la Salud (CUCS), Universidad de Guadalajara, and signed an informed consent statement. Eligibility criteria included adults of 18 years old or older and non-pregnant women. The subjects were classified into two groups: (1) individuals vaccinated without prior COVID-19 infection (*n* = 229), and (2) individuals vaccinated with previous COVID-19 infection (*n* = 117). Both groups were matched by age and gender.

Two surveys were applied to all participants to obtain clinical and demographic data, history of SARS-CoV-2 infection, and vaccine-associated side effects; the first one was at the time of the invitation in the study and the other one 14 days after the vaccine application. Peripheral blood was obtained by venous puncture in Vacutainer tubes without anticoagulant for serum collection. Basal samples were obtained within 3 days after the vaccination in individuals with prior COVID-19 and within 3–5 days post-vaccination in individuals without a history of COVID-19. A second sample was collected 21–25 days after the vaccination in both study groups. This study was conducted according to the Declaration of Helsinki and it was approved by the Committee of Ethics and Biosecurity of the CUCS, Universidad de Guadalajara, Mexico (Registry number 21-10).

Individuals with prior COVID-19 had been diagnosed 1–12 months before the study by RT-PCR (reverse transcription-polymerase chain reaction). They were classified as it follows: asymptomatic, for those without physical symptoms of COVID-19; with mild symptoms, for those who manifested fever, cough, malaise, odynophagia, headache, no dyspnea, oxygen saturation (SO_2_) > 94%, and respiratory rate (R.R.) < 20/min; with moderate symptoms, in those with SO_2_ > 94%, dyspnea or radiological lesions (<50% of pulmonary infiltrates), persistent fever associated with risk factors respiratory rate > 20/min; and with severe COVID-19, for those with SO_2_ < 94% (FiO_2_ 0.21) or R.R. > 30/min or PaO_2_/FiO_2_ < 300 or with pulmonary involvement > 50%.

The absence of a history of COVID-19 was corroborated by the detection of anti-SARS-CoV-2 IgG/IgM antibodies.

### 2.2. Detection of IgG/IgM against SARS-CoV-2

The presence of IgG and IgM antibodies was determined by using the kit Certum IgG/IgM Rapid Test™ cassette (Certum Diagnostics, Nuevo León, Mexico). This test is a lateral flow chromatographic immunoassay for differentiated detection of IgG (sensitivity > 99.9%, specificity 98%) and IgM (sensitivity 85%, specificity 96%) antibodies against SARS-CoV-2. This kit reacts to the presence of nucleocapsid (N) and spike (S) proteins. The protocol was performed according to the manufacturer’s instructions.

### 2.3. Quantification of Neutralizing Antibodies

The quantification of neutralizing antibodies was performed with the cPass™ SARS-CoV-2 Neutralization Antibody Detection Kit (GenScript, Piscataway Township, NJ, USA), which is a blocking Enzyme-Linked Immunosorbent Assay (ELISA). This kit has a 30% signal inhibition cut-off for SARS-CoV-2 neutralizing antibody detection. The neutralization test was performed according to the manufacturer’s instructions. Inhibition rate was calculated as follows:(1)$$\%\;signal\;inhibition=\left(1-\frac{OD\;value\;of\;Sample}{OD\;value\;of\;Negative\;Control}\right)\times 100\%.$$

### 2.4. Quantification of Antibodies against Ad5

The quantification of anti-Ad5 IgG antibodies was performed using the Human Adenovirus Ad5 IgG (ADV-Ad5 IgG) ELISA Kit (MyBioSource, San Diego, CA, USA) that consists of a double antigen sandwich ELISA technique. This assay is based on the features of the target antibody that contains two available paratopes that can be identified by both the pre-coated capture antigen and the detection antigen simultaneously. The detection range for this kit is 1.56–100 ng/mL, with a sensitivity of 0.5 ng/mL. The assay was performed according to the manufacturer’s instructions.

### 2.5. Statistical Analysis

Statistical analysis was performed using the GraphPad Prism v. 6.01 software. The significance level was set at *p* < 0.05. For simple comparisons, we used the Fisher exact test or Student’s *t*-test. Data with nonparametric distribution were represented as median with interquartile range (IQR). For the analysis of variance, the Mann–Whitney U-test was applied for comparing two groups or Kruskal–Wallis for three or more. For comparing two groups of continuous values, we used the nonparametric Wilcoxon signed-rank test. Correlations were evaluated with Spearman’s correlation.

## 3. Results

### 3.1. Description of Study Groups

The clinical and demographic characteristics of both study groups are shown in Table 1. Both groups had similar ages, gender, and comorbidities. Regarding treatment, antidepressants use was more prevalent in the group without prior COVID-19 (*p* = 0.005).

### 3.2. Vaccine-Associated Side Effects

Table 2 shows the side effects associated with the Ad5-nCoV vaccine in individuals with and without prior COVID-19 disease. In both groups, most individuals had side effects (>69%) within the three days after vaccination: headache, myalgia, fatigue, shivers, fever, and arthralgia were the most common effects in both groups. Odynophagia was more prevalent in the individuals with prior COVID-19 than those without prior COVID-19 (*p* = 0.02). Furthermore, we observed a global negative correlation between age and the number of side effects, with the young people who presented more symptoms (r = −0.21, *p* < 0.0001, data not shown). Severe adverse events were not reported in any group. Adverse effects were classified based on the Official Mexican Standard NOM-220-SSA1-2016, Installation and operation of pharmacovigilance.

### 3.3. Generation of Neutralizing Antibodies in Response to the Ad5-nCoV Vaccine

The neutralizing antibodies against SARS-CoV-2 were determined 21–25 days after vaccination in both study groups (Figure 1A,B). A statistically significant increase in the percentage of neutralization was observed after immunization with the Ad5-nCoV vaccine (*p* < 0.0001).

In the group of prior COVID-19 disease (Figure 1B), 100% of the individuals had neutralizing antibodies (>30% of signal inhibition) 21–25 days after the vaccination; however, 7.4% (17/229) of individuals without prior COVID-19 did not have neutralizing antibodies after the same period.

Regarding the basal status, 94% of individuals of the group with prior COVID-19 had neutralizing antibodies against SARS-CoV-2; on the contrary, none of those in the group without prior COVID-19 had these antibodies.

When analyzing the differences in the percentage of neutralization between the two study groups (Figure 2), we observed that individuals with prior COVID-19 had a higher neutralization percentage than those without prior COVID-19 after 21–25 days of vaccination (median [IQR]: 98% [97–98.1] vs. 72% [54–90], respectively; *p* < 0.0001). Surprisingly, the basal neutralization percentage of individuals with prior COVID-19 (before vaccination effects) was higher than those without prior COVID-19 after 21–25 days of vaccination (*p* < 0.001).

The neutralization percentages of the antibodies generated in response to prior SARS-CoV-2 infection or the Ad5-nCoV immunization were also compared among individuals who have had mild, moderate, severe, or asymptomatic COVID-19 illness. Figure 3 shows that individuals with moderate illness presented higher neutralization percentages than those asymptomatic (*p* = 0.012) after natural infection (basal status, Figure 3A). After vaccination, we did not find significant differences among groups (Figure 3B). The correlation between the days after the last COVID-19 positive PCR test and the percentage of neutralization was also evaluated (Figure 3C,D); we observed a negative correlation between both variables; however, this was statistically significant only after 21 days of vaccination (Figure 3D, rho = −0.21, *p* = 0.03).

### 3.4. Correlation between the Percentage of Neutralization with the Clinical and Demographic Variables

We carried out a general analysis (joining both study groups) to identify variables associated with a higher or lower percentage of neutralization. Table 3 summarizes the variables with significant associations. Moreover, Figure 4 shows a negative correlation observed between ages and the percentage of neutralization in individuals without prior COVID-19 after 21 days of vaccination (Figure 4A) and in individuals with prior COVID-19 in basal status (Figure 4B) or after 21 days of vaccination (Figure 4C).

### 3.5. Anti-Ad5 Antibodies and Percentage of Neutralization

The relationship between the percentage of neutralization of the antibodies against SARS-CoV-2 and the anti-Ad5 antibodies levels was examined (Figure 5). A tendency of a negative correlation between both variables was observed, but this was not statistically significant (Figure 5A–C). However, we observed that the anti-Ad5 antibodies levels increased slightly after 21 days of vaccination in individuals without (mean ± SD = 11.2 ± 4.8 ng/mL vs. 12.9 ± 6.4 ng/mL, *p* = 0.003, Figure 5D) or with prior COVID-19 (mean ± SD = 9.9 ± 2.1 ng/mL vs. 12.9 ± 1.7 ng/mL *p* = 0.005, Figure 5E).

## 4. Discussion

The current COVID-19 pandemic has tested the ability to produce vaccines in a record period. In the absence of a specific treatment for COVID-19, it is essential to reduce the incidence of cases and the mortality rate; therefore, vaccination programs became a priority in all countries. Some of them, like Mexico, are even testing vaccines that have not yet completed phase III trials.

So far, there are only phase I and II studies of the Ad5-nCoV vaccine, for this reason, there are many uncertainties in the Mexican population about its efficacy and safety. This study focused on analyzing the differences in the neutralizing antibodies production induced by the Ad5-nCoV vaccine (commercial name “Convidicea”) in individuals with and without a history of COVID-19, as well as identifying potential clinical factors that could modify them.

In contrast with other studies [18,19,20], we did not find a higher prevalence of comorbidities (diabetes, obesity, or SAH) in individuals with prior COVID-19 than those without prior COVID-19. This could be attributed to the high prevalence of these comorbidities in the general Mexican population [21].

The analysis of the pharmacological treatment showed a higher antidepressant consumption in the group without prior COVID-19. The COVID-19 pandemic has led to an increase in the incidence of depression, so the use of antidepressants has increased, and this has been surprisingly linked to protection against SARS-CoV-2 [22,23]. The mechanism of this association is unclear, but it could be suspected that some antidepressants functionally inhibit acid sphingomyelinase activity (ASM), and in cell culture models, inhibition of ASM prevents the infection of cells with SARS-CoV-2 [24]. Moreover, antidepressant use has been associated with a reduced risk of intubation or death in hospitalized patients with COVID-19 because the inhibition of ASM decreases pro-inflammatory mediators such as IL-2, IL-6, IL-7, IL-10, TNF-α, C-reactive protein (CRP), and D-dimer [22,25,26].

Overall, we observed that the Ad5-nCoV vaccine has good tolerance in both groups, presenting reactogenicity with mild to moderate symptoms that did not require additional medical attention. In both study groups, the major side effects reported up to three days after vaccination were fever, fatigue, headache, myalgia, and arthralgia; these symptoms are concordant with the reported by Zhu et al., in phase I trial of the Ad5-nCoV vaccine, only with the difference that the side effects occurred up to seven days later [27]. Phase II of this vaccine shows the presence of the same secondary symptoms [11]. Other vector vaccines such as ChAdOx1 nCoV-19 (Oxford- AstraZeneca) [28] and Gam-COVID-Vac (Sputnik V) [29], showed similar side effects of those observed in the present study.

The presence of odynophagia was a vaccine side-effect more prevalent in individuals with prior COVID-19; however, this result should be interpreted with caution due to the low number of individuals who presented it in both study groups. Zhu et al. also reported this symptom without any clinical relevance, but in a study of the ChAdOx1 nCoV-19 vaccine, this side-effect was classified as a reaction similar to allergies [30].

Another interesting observation from our study was a negative correlation between age and the number of vaccine side effects, a finding that has not been reported for this vaccine; however, in other vector vaccines such as ChAdOx1, a higher frequency of side effects in young people than in older people has been reported. This could be explained by taking into consideration that young people have a more robust immune system [31] while older adults have immunosenescence [32].

Regarding the production of antibodies with neutralizing capacity, a significant increase in the percentage of signal inhibition was observed in both study groups after vaccination, which is consistent with those reported in other studies [13,33]. Zhu et al. reported that patients in phase I and II trials of Ad5-nCoV begin to produce neutralizing antibodies on day 14 and peaked 28 days post-vaccination [11,27]. However, our results show that in the group without prior COVID-19, 17 individuals (7.4%) did not have the presence of neutralizing antibodies after vaccination, which is contradictory to previous reports [11,27]. However, the ChAdOx1 vaccine reported negative cases at its first dose, and after the second dose, all subjects had neutralizing antibodies [34]; unfortunately, Ad5-nCoV is currently applied in a single dose, so we suggest reviewing that recommendation.

Among the 17 seronegative subjects for neutralizing antibodies against SARS-CoV-2, three had an autoimmune disease, and we can attribute the lack of neutralizing antibody generation to their immunosuppressive treatment [35]. For the rest of the cases, we did not observe any apparent variable that could explain this finding, therefore, it may be directly due to the immune response of each patient, where genetic factors or nutritional status may interfere with the response to vaccination [36].

For the group with prior COVID-19, all patients had antibodies with neutralizing capacity 21 days post-vaccination. This could be explained by the presence of neutralizing antibodies in 94% of these individuals before vaccination, by natural infection. Therefore, a vaccination enhances and optimizes the neutralization capacity in these individuals, and a single dose of Ad5-nCoV could be sufficient for this group, as proposed for other vaccines [37,38,39,40].

Surprisingly, we observed that the basal neutralization percentage (before vaccination) of the group with prior COVID-19 was higher than the vaccinated individuals with no history of COVID-19. Therefore, we suggest that the application of a next dose of this vaccine is required, especially for individuals without previous COVID-19. This is an issue that must be verified because the baseline levels of patients with previous COVID-19 were determined within the first three days after vaccination and although it is unlikely, a low number of antibodies could have been rapidly generated due to the pre-existing B cell memory.

We also observed an association between severe COVID-19 clinical course and higher percentage of neutralizing antibodies (basal percentage) after a natural infection, which is consistent with previous reports [41,42]. However, after vaccination, we did not observe this difference, which differs from other studies [43,44].

It is important to determine how long neutralizing antibodies remain active; we observed the presence of neutralizing antibodies more than 300 days after COVID-19 diagnosis; however, the neutralization percentage tends to decrease after this time, which affects the neutralization percentage post-vaccination. These findings support the importance of vaccine administration in people with a history of SARS-CoV-2 infection, which indeed, generates a greater humoral response against SARS-CoV-2 [43].

In the present study, we also observed potential factors that could affect the percentage of neutralization; however, these associations should be confirmed in future studies as the analyses were performed in small subgroups of individuals. One of those associated factors was the use of immunosuppressants (glucocorticoids and rituximab) or antidepressants. These drugs, despite helping in the prevention of complications of COVID-19, could be associated with low neutralizing antibodies percentage due to their inhibitory effects on cytokines, some of which are important for humoral response [22,45]. Previous studies that used antidepressants drugs as preventive treatments [22,25,26] have not reported similar results to us. We consider this topic interesting due to increase of mental illness associated with the COVID-19 pandemic [23].

Regarding immunosuppressants, a previous study showed that most of patients using rituximab were seronegative for anti-SARS-CoV-2 antibodies after-vaccination [46]. Another study showed that patients treated with glucocorticoids exhibit impaired SARS-CoV-2 vaccine-induced immunity, with a reduction of humoral response [47].

Otherwise, the presence of side effects such as fever, shivers, and arthralgia were statistically related to the production of antibodies. This can be attributed to the effects of some cytokines, such as interferon type I (IFN-I), that has a high production in the early stages of the viral immune response, and that is part of the innate immunity [48]. This association is consistent with other publications [44,49].

A clinical variable associated with a lower neutralizing antibodies percentage in both study groups (with and without prior COVID-19) was age, which suggests the presence of an immunosenescence phenomenon [50,51,52].

The main disadvantage of viral vector-based vaccines is the presence of pre-existing immunity against the vector [53]. In this study, we did not find a correlation between antibodies against adenovirus 5 (Ad5) and the neutralizing antibodies for SARS-CoV-2, which is consistent with the pilot study of the Ad5-nCoV vaccine [27]. In consequence, we ruled out that the 17 seronegative cases for neutralizing antibodies against SARS-CoV-2 could be related to the previous existence of antibodies against Ad5. However, in both study groups, it was observed that the levels of antibodies against Ad5 increase slightly 21-days post-vaccination; therefore, this finding must be replicated in a larger longitudinal study and taken into consideration if a booster dose of this vaccine is established.

Despite the fact that our results did not show a correlation of anti-Ad5 antibody levels with the neutralization percentage of anti-SARS-CoV-2 antibodies, phase I and II clinical studies of Ad5-nCoV vaccine reported a high prevalence of these antibodies, and after immunization, a marked increase (four times) above baseline value was observed [11,27]. A limitation of our study is the lack of positive control to Ad5 seropositivity, thus, not being able to cut positive and negative values of the presence of antibodies against Ad5. Also, another limitation is the lack of epidemiological studies about the seroprevalence of these antibodies in our country.

The company CanSino has proposed a booster dose after six months, this could be possible as long as the dose is high enough to overcome the effect of antibodies generated against the vector, or another option is to look for another non-human adenovirus vector [54].

A limitation of this study is that we do not have quantitative determinations to make the correlation between antibody levels and their neutralizing capacity. However, our competitive ELISA plate for measuring the blocking of neutralizing antibodies is equivalent to the standard gold neutralization test (VNT) [55,56]. Studies such as that of Sahin et al. did not make any difference in comparing these two methodologies [57].

A strength of our study is the comparison of the effect of the vaccine in two large groups, in those with and without history of COVID-19, which contributes to the insight in the production of neutralizing antibodies, which is not considered in the CanSino’s phase I–III clinical studies.

## 5. Conclusions

In conclusion, our results showed a higher neutralization percentage against SARS-CoV-2 after immunization with Ad5-nCoV, in individuals with prior COVID-19 disease. This finding could be useful to define criteria for the application of this vaccine in countries such as Mexico. We also showed that age, pharmacological treatments such as antidepressants and immunosuppressants, and the presence of reactogenicity to vaccination are potential factors associated to a positive or negative outcome response to vaccination.

Finally, anti-Ad5 antibodies induction after vaccination is a potential risk that must be considered and evaluated before deciding to use a booster with the same Ad5-nCoV vaccine.

## Figures and Tables

**Figure 1 vaccines-09-01047-f001:**
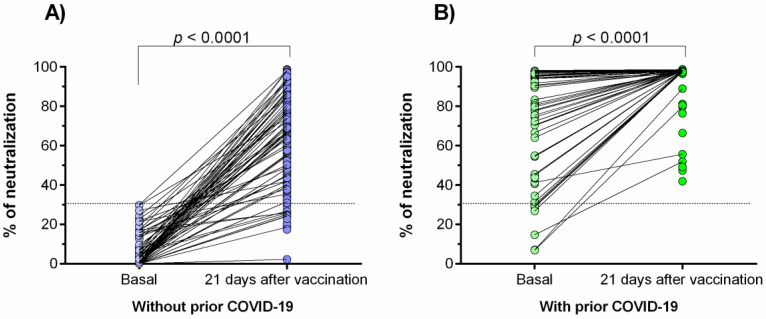
Percentage of neutralizing signal of antibodies generated in response to the Ad5-nCoV vaccine. (**A**) Individuals without prior COVID-19; (**B**) individuals with prior COVID-19. Differences were calculated by the Wilcoxon signed-rank test. The dotted line indicates the cut-off point for the neutralization test (>30%). The “basal” status represents the neutralization percentage before the effect of vaccination (3 days after vaccination).

**Figure 2 vaccines-09-01047-f002:**
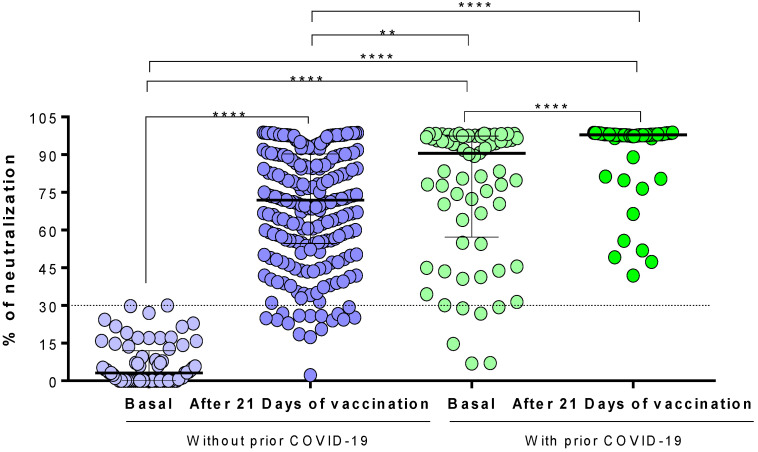
Comparison of the neutralization percentages of the antibodies generated in response to the Ad5-nCoV vaccine. The difference between all groups was calculated with the U-Mann–Whitney test or the Wilcoxon signed-rank test. The data are provided as medians and interquartile ranges. ****, *p* < 0.0001; **, *p* < 0.001.

**Figure 3 vaccines-09-01047-f003:**
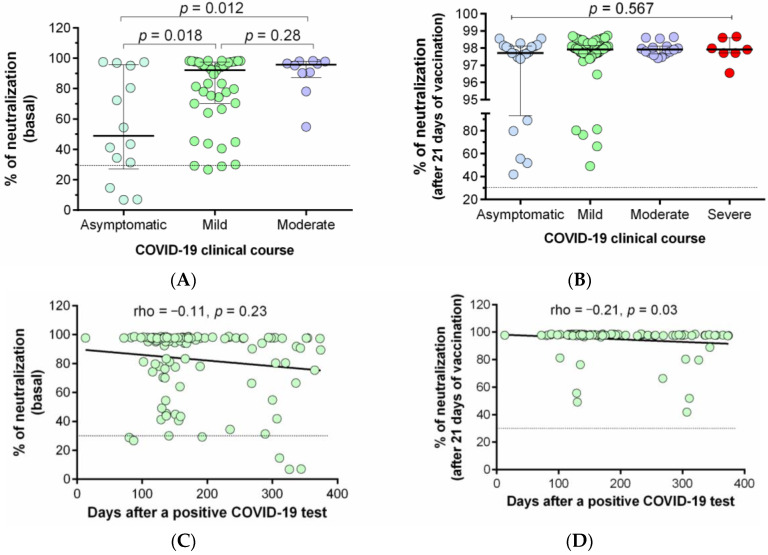
Relationship between prior COVID-19 clinical course and days after infection with the percentage of neutralization. (**A**) Comparison of basal neutralization percentage between individuals with different prior COVID-19 clinical courses; (**B**) comparison of neutralization percentage after 21 days of immunization with the Ad5-nCoV vaccine among individuals with different prior COVID-19 clinical courses. Correlation between percentage of neutralization and the time of a previous COVID-19 infection: (**C**) Basal neutralization percentage, and (**D**) neutralization percentage after 21 days of the vaccination. The difference between all groups was calculated with the Kruskal–Wallis test, followed by the U-Mann–Whitney test. Data are provided as median and interquartile ranges.

**Figure 4 vaccines-09-01047-f004:**
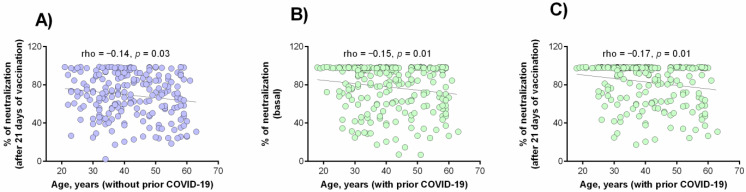
Correlation between the age and neutralization percentage. (**A**) After 21 days of vaccination in patients without prior COVID-19; (**B**) in basal status in patients with prior COVID-19; (**C**) after 21 days of vaccination in patients with prior COVID-19. Analysis was evaluated by the Spearman’s rank correlation coefficient.

**Figure 5 vaccines-09-01047-f005:**
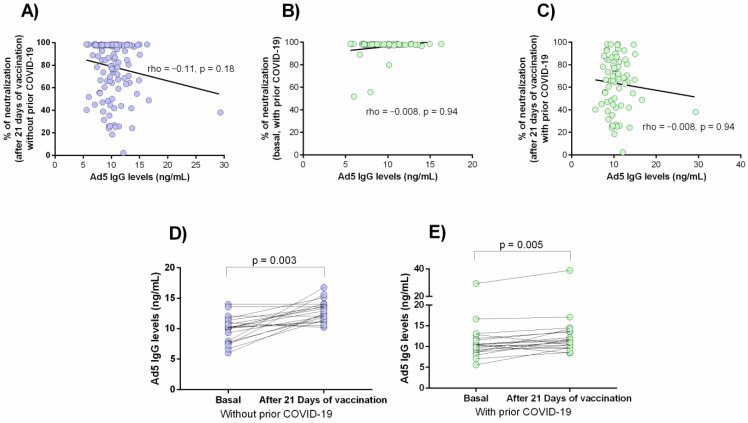
Anti-Ad5 antibodies and percentage of neutralization. Correlation between anti-Ad5 antibodies and percentage of neutralization of anti-SARS-CoV-2 antibodies: (**A**) after 21 days of vaccination in patients without prior COVID-19, (**B**) in basal status in patients with prior COVID-19, and (**C**) after 21 days of vaccination in patients with prior COVID-19. Levels of anti-Ad5 antibodies generated in response to the Ad5-nCoV vaccine in individuals without (**D**) and with (**E**) prior COVID-19.

**Table 1 vaccines-09-01047-t001:** Clinical and demographic characteristics of the study groups.

	Immunized with Ad5-nCoV Vaccine		*p*-Value
	without Prior COVID-19 *n* = 229	with Prior COVID-19 *n* = 117	
**Age (years)**, ^mean±SD^	41.6 ± 10.6	39.8 ± 10.9	0.128
Gender, *^n^*^,(%)^			
Female	166 (72.5)	79 (67.5)	0.338
Male	63 (27.5)	38 (32.5)	
Days of the last positive PCR test for COVID-19 ^median(IQR)^	-	158 (130–241)	-
**Comorbidities**, *^n^*^,(%)^			
None	101 (44.1)	50 (42.7)	0.80
Overweight/obesity	75 (32.8)	41 (35.0)	0.67
Allergic diseases	32 (14.0)	13 (11.1)	0.45
SAH	24 (10.5)	9 (7.7)	0.39
Diabetes	15 (6.6)	5 (4.3)	0.39
Hypothyroidism	8 (3.5)	2 (1.7)	0.35
Autoimmune diseases	8 (3.5)	1 (0.9)	0.14
Dermatitis	3 (1.3)	3 (2.6)	0.39
Dyslipidemia	4 (1.7)	2 (1.7)	0.98
Heart diseases	4 (1.7)	1 (0.9)	0.51
**Treatment**, *^n^*^,(%)^			
Antihypertensive	25 (10.9)	10 (8.5)	0.49
Antidepressants	23 (10.0)	2 (1.7)	0.005
Hormonal	20 (8.7)	7 (6.0)	0.368
Hypoglycemic agents	17 (7.4)	6 (5.1)	0.41
Hypolipidemic agents	3 (1.3)	4 (3.4)	0.18
NSAIDs	7 (3.1)	3 (2.6)	0.79
Antihistamines	3 (1.3)	2 (1.7)	0.76
Immunosuppressants	4 (1.7)	1 (1.7)	0.51

SD, standard deviation; SAH, Systemic arterial hypertension; NSAIDs, Non-steroidal anti-inflammatory drugs; *p*-values were calculated by Fisher’s exact (*^n^*^,(%)^) or *t*-student (^mean±SD^).

**Table 2 vaccines-09-01047-t002:** Side effects to Ad5-nCoV vaccine.

Side Effects	Immunized with Vaccine Ad5-nCoV		*p*-Value
	without Prior COVID-19 *n* = 229	with Prior COVID-19 *n* = 117	
**At least one** ^median(Q25–Q75)^	160 (69.9)	86 (73.5)	0.482
Number of symptoms, ^mean±SD^	2.28 ± 2.321	2.85 ± 2.964	0.054
Headache *^n^*^,(%)^	107 (46.7)	58 (49.6)	0.617
Myalgia *^n^*^,(%)^	82 (35.8)	52 (44.4)	0.119
Fatigue *^n^*^,(%)^	92 (40.2)	48 (41)	0.879
Shivers *^n^*^,(%)^	54 (23.6)	34 (29.1)	0.270
Fever *^n^*^,(%)^	48 (21)	33 (28.2)	0.133
Arthralgia *^n^*^,(%)^	45 (19.7)	33 (28.2)	0.072
Irritability *^n^*^,(%)^	29 (12.7)	13 (11.1)	0.677
Abdominal pain *^n^*^,(%)^	11 (4.8)	10 (8.5)	0.169
Odynophagia *^n^*^,(%)^	8 (3.5)	10 (8.5)	0.002
Rhinorrhea *^n^*^,(%)^	15 (6.6)	6 (5.1)	0.601
Cough *^n^*^,(%)^	9 (3.9)	3 (2.6)	0.513
Dizziness *^n^*^,(%)^	6 (2.6)	3 (2.6)	0.975
Conjunctivitis *^n^*^,(%)^	4 (1.7)	3 (2.6)	0.611
Application-site pain *^n^*^,(%)^	3 (1.3)	3 (2.6)	0.399
Vomit *^n^*^,(%)^	5 (2.2)	2 (1.7)	0.768

SD, standard deviation. *p*-values were calculated by Fisher’s exact (*^n^*^,(%)^), *t*-student test (^mean±SD^), or the U-Mann–Whitney test (^median(Q25–Q75^^)^).

**Table 3 vaccines-09-01047-t003:** Factors associated with the percentage of neutralization after vaccination.

	Percentage of Neutralization, Median (Q25–Q75) (*n* = 346)	*p*-Value
**Comorbidity**		
With autoimmunity (*n* = 9)	60.58 (24.98–88.85)	0.053
Without autoimmunity (*n* = 337)	87.90 (62.49–97.79)	
**Treatment**		
Use of antidepressants (*n* = 25)	72.65 (39.43–87.77)	0.017
No use of antidepressants (*n* = 321)	89.61 (61.64–97.81)	
Use of immunosuppressants (*n* = 5)	23.52 (22.85–28.5)	0.018
No use of immunosuppressants (*n* = 341)	86.76 (61.16–97.78)	
**Side-effects to Ad5-nCoV vaccine**		
With fever (*n* = 81)	96.20 (73.47–97.96)	0.008
Without fever (*n* = 265)	83.60 (56.51–97.75)	
With shivers (*n* = 88)	91.03 (73.96–97.94)	0.038
Without shivers (*n* = 258)	84.41 (57.08–97.76)	
With arthralgia (*n* = 78)	95.79 (73.34–97.99)	0.009
Without arthralgia (*n* = 268)	83.60 (58.29–97.72)	

*p*-values were calculated by the U-Mann–Whitney test.

## Data Availability

The data that support the findings of this study are available on request from the corresponding author.

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
