# Peer review of "Neutralizing Antibodies against SARS-CoV-2, Anti-Ad5 Antibodies, and Reactogenicity in Response to Ad5-nCoV (CanSino Biologics) Vaccine in Individuals with and without Prior SARS-CoV-2"

_vaccines, 2021, doi:10.3390/vaccines9091047_

Round 1
Reviewer 1 Report
Reviewer’s Comments - Manuscript Vaccines-1360002 “Neutralizing Antibodies against SARS-CoV-2, anti-Ad5 antibodies, and reactogenicity in response to Ad5-nCoV (CanSino Biologics) vaccine in individuals with and without prior SARS-CoV-2” by Jorge Hernández-Bello et al.
In the interesting manuscript submitted by Hernández-Bello et al., the Authors evaluate the generation of SARS-Cov-2 neutralizing antibodies and factors associated with the degree of immune response elicited by a single dose of the Ad5-nCoV vaccine (by CanSino Biologics) in subjects who had, and had not, experienced previous COVID-19. The reported results indicate that i) the vaccine induced significantly higher percentage of neutralizing antibodies in individuals with prior COVID-19 than those without prior COVID-19, ii) natural infection before vaccination induced higher percentage of neutralizing antibodies than immunized individuals without prior COVID-19, iii) the age and the use of antidepressant and immunosuppressive treatments, reactogenicity to the vaccine, and history of COVID-19 are associated with impaired antibody production. Finally, anti-Ad5 antibodies increased significantly three weeks after vaccination in all groups.
In my opinion, the message of this manuscript is important as it is an independent study outside of clinical trials (phase I-III) and addresses relevant issues to the field of COVID-19 vaccine development i.e. factors affecting the immune responses to the vaccine, the induction of Ad5 antibodies that must be considered if given a second dose and the entity of neutralizing antibody response depending or not on the presence of pre-existing immunity due to a previous infection and depending also on the severity of the clinical disease. The study certainly fits the scope of the journal and the topic may be of interest to the vast readership of Vaccines Journal and more specifically to researchers working in the field of Sars-CoV-2 research and vaccine development.
line 345, I believe that one [53] citation should be deleted.
I do not have other specific comments.
Reviewer 2 Report
This study, showing the ability of the Ad5-nCoV vaccine to generate neutralizing antibodies in individuals with and without prior COVID-19 disease, without causing significant adverse effects, clearly supports the notion of administering a booster dose of this vaccine especially to individuals without previous COVID-19 infection. It is a very well and clearly written article in which the conclusions are supported by the data presented. I recommend publishing the article after revising it according to the following minor suggestions.
- Line 54: It is mentioned that Ad5-nCoV is currently undergoing Phase III clinical evaluation, and reference 9 is given. However, the clinical trial referenced is a Phase IIb. Please clarify or correct.
- Reference 10: It says that it was “accedido”, which is in Spanish– should be accessed.
- Line 57: Should be 5×1010 viral particles per mL.
- Line 75: Reference 15 does not seem relevant to the statement it refers to. Please replace with a more appropriate reference.
- Lines 199, 230, 256 and 338: The expression “on the other hand” should be used when in a previous statement the expression “on one hand” has been used. Suggest revising the previous sentences accordingly, or use a different expression.
- Line 292: Suggest “17 individuals” rather than “17 people”.
- Line 321: “tends to decrease”.
- Line 323: should be “indeed”.
- Line 331: “have not reported”.
Reviewer 3 Report
Major comments:
- the baseline samples is taken already 3-5 days after immunization. while probably of minimal impact in subjects without prior COVID-19 exposure, it can be excluded (and it is even likely) that at this time point a memory response could already be observed in subjects with prior COVID-19 exposure. therefore, the comparison of basal level in this population to the post immunization titers in subjects without prior exposure is not valid and the conclusion that a booster is required on line 313 is too strong and not fully supported by the data.
- many statistical comparisons were performed, often on small numbers per subgroup. it is not clear from the methods if these analyses were pre-specified or post-hoc. if these analyses were exploratory or post hoc, the conclusions are too strong for the robustness of the data, as exemplified by many overlapping confidence interval between the groups.
Round 2
Reviewer 3 Report
I appreciate the effort of the authors to take the recommendations into account. the caveats of the research - inherent to these types of clinical studies - are now well covered in the manuscript.